# Indexes for the E-Baking Tray Task: A Look on Laterality, Verticality and Quality of Exploration

**DOI:** 10.3390/brainsci12030401

**Published:** 2022-03-17

**Authors:** Antonietta Argiuolo, Federica Somma, Paolo Bartolomeo, Onofrio Gigliotta, Michela Ponticorvo

**Affiliations:** 1Natural and Artificial Cognition Laboratory, Department of Humanistic Studies, University of Naples Federico II, 80133 Naples, Italy; federica.somma@unina.it (F.S.); onofrio.gigliotta@unina.it (O.G.); michela.ponticorvo@unina.it (M.P.); 2Institut du Cerveau—Paris Brain Institute—ICM, Inserm, Hôpital de la Pitié Salpêtrière, Sorbonne Université, 75013 Paris, France; paolo.bartolomeo@icm-institute.org

**Keywords:** E-BTT, Baking Tray Task, neglect, indexes, spatial cognition, assessment

## Abstract

The Baking Tray Task is an ecological task developed for the assessment of unilateral neglect that can also be used for research on neurotypical participants. In this task, participants are asked to place 16 objects inside a board as evenly as possible. In the case of impaired spatial exploration, consequent to right attentional networks damage, asymmetrical object disposition is observed as more objects are placed on the ipsilesional side (typically the right side). The E-BTT is a technology-enhanced version of the Baking Tray Task, implemented with a software platform, E-TAN, which detects the objects and automatically computes their spatial coordinates. This allows a complement to the traditional scoring methods with new measures to extract richer information from the data. In this study, we focus on neurotypical participants to explore if some new indexes, derived from the literature review on similar tasks, can be applied to BTT and E-BTT for research aims. A principal component analysis (PCA) was then performed to verify if these new indexes reflect some common dimensions. Results indicate the emergence of two principal dimensions: spatiality, which summarizes both laterality and verticality, and quality, which regards the explored space and (dis)organization in placing the items.

## 1. Introduction

The assessment of unilateral neglect takes advantages of a large number of tools. One of these, the Baking Tray Task [1], is an ecological test that consists of placing sixteen 3.5 cm cubes into a 100 × 75 cm black board with an edge of 3.5 cm in height. The metaphor deals with putting, as evenly as possible, the 16 “buns” within the “tray” as if they were to be baked in the oven. Given that this task refers to a simple daily activity, such as baking, it was considered an ecological alternative to pencil-and-paper tasks.

Since the Baking Tray Task was developed as a neglect assessment tool, the original sample consisted in right brain-damaged patients. Compared with a control group of healthy subjects, they placed the cubes in a right-side-unbalanced way. That meant placing most cubes on the right half of the board, reflecting an impaired spatial exploration. On the other hand, control subjects placed the cubes evenly among the board, usually symmetrically (eight on each side). Their worst performance (an asymmetry of more than two cubes) was then considered as a cutoff for neglect assessment [1].

Along with diagnostic and clinical goals, these tasks can be used in the research on spatial attention to verify if spatial exploration is symmetrical [2].

In recent years, the use of supports such as a PC or tablet extended to neuropsychological assessment, leading to the digitization of new tools and new versions of already validated tests. This brought several advantages in testing and scoring procedures. Furthermore, digital supports allow scholars to have information that is difficult to collect in other ways.

As with other tests, the Baking Tray Task was also the object of a technological advance using several devices [3,4,5,6]. Cubes were transformed into tangible interfaces by applying ArUco Marker Tags on them [7]. Tags were also used to digitally assess the available area, which was possible thanks to the application of a software tool, E-TAN (Figure 1). The board and the cubes were replaced by a wooden modular frame and a series of 16 disks.

The E-TAN platform, thanks to the application of a Logitech camera, has the capacity to detect the four tags in the corners of the frame and can calculate the available area within which the disks will then be arranged. On each of them, other tags are also affixed (a different one on each disk to identify the single ID) from the same library developed by Garrido-Jurado and colleagues [7]. Then, for each identified disk, the software records the timestamp and a pair of coordinates (X and Y). The coordinates and the area are measured in pixels; the center (0.0) coincides with the lower right corner. In recent studies [2], it was decided to change the reference system so that the center would fall in the real center of the frame and express the coordinates in centimeters. The choice of centimeters instead of pixels was motivated by the convenience of using the same unit of measure as the board. Moreover, changing the center permits positive X coordinates when referring to the right and negative to the left; similarly, Y coordinates are positive when the disk is placed in the upper side and negative when it is down.

The enhanced version of the Baking Tray Task (called from now on E-BTT) was recently applied in some studies [3,4,5,6], but it needs further validation, given that we are dealing with 16 coordinates. One of its strengths, though, could be considered its ecological validity as in the original version by Tham and Tégner.

Traditional scoring methods, derived from the original BTT, consist of counting how many cubes/disks were placed on each side. A difference greater than two was considered a sign of neglect [1,3,8,9]. A more refined formula was developed by Facchin and colleagues [10] to measure the percentage of right/left bias. It is a percentage given by multiplying by 100 the ratio between the right/left unbalance and the total number of placed cubes. The right/left difference and the laterality bias were applied to the E-BTT [4]. In addition to that, Cerrato and colleagues, who first developed the prototype of E-BTT, performed a quadrant analysis [4], dividing the space in four equal parts. The first and the last disks’ positions were studied to discover preference in starting or finishing the spatial exploration. The majority of healthy participants started from the top part and finished in the bottom right [4]; this evidence was confirmed in later studies [6].

In another study, the convex hull area described by the disks, taken as vertices of a polygon, was also considered [3]. This was regarded as an estimation for each participant or patient of the processed portion of space. The results showed that this index could actually discriminate neglect from non-neglect patients.

The E-BTT was also used in a recent work on pseudoneglect [2]. Using the first disk’s X coordinate and the mean of all 16 disks’ coordinates, Somma and colleagues proved that in E-BTT, the spatial exploration started slightly on the left and the center of mass (mean of all the X coordinates) was there shifted.

Another study used Euclidean distances applied to E-BTT coordinates, also considering the temporal order of placement. This attempt resulted into two different measures [11]: between distance (BD) and within distance (WD). The between distance (BD) is the sum of the distances between a disk of each participant and the corresponding disk of another participant. Thanks to this kind of comparison, eleven groups of strategies emerged in neurotypical participants. The within distance (WD), instead, is the sum of the distances between a disk and the next one within the same sequence; it is, thus, larger in spatially disorganized patterns.

To summarize, the digital enhancement of the Baking Tray Task establishes the disks coordinates and their spatio-temporal sequence that can be useful to determine the strategies the participant uses to solve this particular spatial task. In order to deepen the understanding of these different strategies, we identified and applied new indexes on the data from E-BTT with the twofold goal of getting richer information from the data offered by E-BTT (namely, spatio-temporal sequences) and to make a first attempt to register normative data for these indexes.

In this paper, we propose, in more detail, the application of several indexes to the E-BTT.

These indexes result from the following sources:already used indexes for the E-BTT (total area, first X, etc.) [3,6];index from the visual search organization literature (best R, intersecion rate, etc). They estimate the efficiency of spatial organization in item cancellation tasks [12,13,14,15];new indexes (e.g., between distance from optimal sequences) [11]. These indexes were developed specifically for E-BTT data and can be generalized to all tasks that involves coordinates.

Starting from these indexes, our purpose was to understand whether these indexes reflect common dimensions and whether these features can be related to aspects of spatial attention and peripersonal exploration, firstly in a nonclinical population. For this reason, in this paper, we will focus on healthy participants.

## 2. Materials and Methods

### 2.1. Participants

The sample consists of 122 healthy participants (97 female), whose age ranged from 18 to 37 years old (mean age = 21.33, SD = 3.84) even though we are aware that the small number of participants is not adequate for a validation. This choice was motivated by the fact that the E-BTT can be applied both on healthy and impaired participants. Some of the data were used in a study on pseudoneglect [2]. The present study was approved by the University of Naples “Federico II” Local Ethics Committee and was conducted in accordance with the Declaration of Helsinki. Informed consent was obtained by all participants.

### 2.2. Measures

#### 2.2.1. Enhanced Baking Tray Task

The task was administered following Tham and Tegnér’s original procedure, with some differences due to the technological enhancement. In its latest version, the E-BTT is made up of the following (Figure 1):A 60 × 45 cm modular wooden frame, 5 cm wide on each side. This was different from the first board used in [1] because a smaller area was preferred. Tham and Tegnér used a smaller area as well and found overlapping results in respect to the bigger tray [1].Cubes were replaced by disks because the camera worked better with flat objects; moreover, cubes were subject to an aggregation bias (that is, putting all cubes close to form another object). Disks measured 5 cm in diameter.ArUco marker tags on each disk and frame’s corner [7]. They consisted of a black-and-white matrix, similar to QrCodes.A Logitech C930e webcam camera placed above the table, fixed thanks to a metallic arm in order to facilitate object detection.E-TAN, the software part of the E-BTT. It is a versatile platform developed ad hoc for the E-BTT but can potentially be applied to many other tasks. It allows tangible interfaces detection inside the “tray” and calculates many important variables, such as the coordinates of each detected disk, along with its time stamp. Moreover, information about each session (board dimension, date, gender and age of the participants) are recorded.

The E-TAN platform was implemented on a personal computer connected to the 30-fps camera (Webcam Logitech C930e; Newark, CA 94560, USA) through a USB cable. The camera was placed above the board thanks to a metallic arm; the distance was regulated to focus on the frame’s edge. The E-BTT was made up of a 60 × 45 cm wooden frame and sixteen 5 cm wooden disks (height 1 cm), both equipped with ArUco marker tags [7]. The long side of the frame was arranged so that it was aligned with the sagittal plane of the participant. The 16 disks were stacked in four piles in the space between the frame and table edge. The participants were asked to place, as evenly as possible, the disks inside the frame as if “they were cookies to be baked”. The only rules to follow were to use one hand, and to not move a disk once it was placed.

#### 2.2.2. Indexes

We calculated the indexes listed in Table 1. According to the way indexes are computed, we can distinguish, in measurement, where (spatiality) or how the disks were placed (quality).

**Spatial indexes.** Spatial indexes summarize where the disks were placed. They are further divided in laterality and verticality indexes, depending on which kind of coordinate they were calculated on (X or Y). The quadrant analysis is classified separately since it comprehends both the vertical and the lateral dimensions.
◦**Quadrant analysis (first and last disk).** The internal space of the frame is virtually divided in four equal parts based on each axis. The disks’ placement frequencies are counted and analysed through a one-way Pearson’s chi-square in order to reveal where it is more likely to place the first or the last disk. It is also possible to compute a two-way chi-square, considering laterality and verticality as two different variables, but for theoretical reasons, it was preferred not to.Quadrant analysis could be helpful and fast to detect tendency in spatial exploration.◦**Laterality indexes.** All indexes measured using the X coordinates were considered measures of laterality. Laterality indexes gave the idea of how much to the right or to the left one disk or the entire sequence was located. Laterality indexes are as follows:
■*First X.* It corresponds to the first placed disk’s X coordinate, in cm. It can be considered an index of pseudoneglect [2]. The rationale behind this was that spatial exploration started asymmetrically, and in fact the first disk was placed leftward.■*Right/left disks’ difference (R/L difference).* The number of disks placed on the left is subtracted from the number of disks placed on the right to obtain an estimate of the imbalance in disks’ placement. A positive number means an asymmetry towards the right while a negative number would mean more disks placed on the left. Zero means symmetry.■*Laterality bias.* The laterality bias (LB) was developed by Facchin and colleagues [10] as the ratio between the right/left cubes difference and their total number, in a percentage. This formula was thoroughly used for coordinate data, considering every left disk had a X coordinate less than −10 and every right disk had a X coordinate more than +10. Facchin and colleagues [10] calculated two cutoff scores using nonparametric tolerance intervals: −12.6% for the left side and +18.8% for the right side.■*Distance lateral gradient.* It was developed by Rabuffetti and colleagues [14]. The gradient was developed to assess possible relationships between test performance and laterality. The slope of the regression line on the intercancellation distance (similar to the “within distance”, see below), putting the lateral coordinate as a predictor, was computed. The slope of the fitting line corresponds to the distance lateral gradient. Please note that only the coordinates from the second one were considered to match the fifteen distances. This index can be interpreted as the variation on the distance between each disk and the next one, moving by one unit (in this case, centimeters) in laterality. A positive gradient means that the more the coordinates vary rightward, the higher the distances will be between each disk and the next one.■*Center of mass (mean_X).* It is calculated as the mean of all 16 coordinates. It gives an idea of how much the overall configuration is biased toward the left or the right. A positive center of mass indicates that the configuration is biased rightward.
◦**Verticality indexes.** Verticality was not taken into account in previous studies [4] because vertical neglect is a relatively rare occurrence [16]. Nonetheless, it should be interesting to assess verticality in healthy participants because the role of verticality in visual search is underexplored. Verticality indexes are analogous to their laterality counterparts: *first Y, up/down disks’ difference (U/D difference), verticality bias, distance vertical gradient, center of mass (mean_Y).*
***Quality indexes.*** Quality indexes refer to the final sequence’s quality and organization. Some of them derive from the visual search literature on cancellation tasks [12,13,14,15]. Visual search organization was initially investigated mainly with patients with unilateral brain injury (especially in the right hemisphere, with or without unilateral neglect), highlighting how they tended to describe irregular exploratory patterns whereas neurologically healthy subjects had a more organized and regular pattern of cancellation [12,15,17,18,19,20]. Mark and colleagues [12], in fact, tried to quantify the spatial organization in a Star Cancellation task in patients who had suffered a stroke with three indexes. Subsequently, other authors [13,14] used the same indexes and formulated new ones, always applying them to the task of cancellation of stimuli. For the E-BTT task, we chose to use the following indexes: number of intersection (intersection rate, longest path), global speed, best R and standardized angle.
■*Total area*. The proportion of explored area was calculated through the Monte Carlo integration algorithm. In particular, the convex hull polygon delimited by the disks’ sequence (for more detail, please refer to [3]) was considered. Since outliers in disks’ placement could alter the estimate, the external and internal polygon were averaged. In Cerrato and colleagues, the log transformation of the portion of explored area was divided into a left and a right portion. This index proved to be useful to discriminate patients from healthy subjects: only 8% of healthy participants had a pathological area. The quantity of occupied space inside the frame could be an index of performance quality.■*Total time*. In this study, performance time was calculated as the time interval between the placement of the final and the first disk. It could be regarded as a (dis)organization index since difficulties of spatial exploration should lead to longer execution times. In cancellation tasks, performance time was used as a sustained attention measure [13].■*Within distance (WD).* Within distance was the sum of the Euclidean distances between each disk and the next one. It can be regarded as the measure of total distance of explored space with the disks. It was recently used as an organization measure in Argiuolo and colleagues [11], but a similar index was previously applied to the cancellation task by several other authors [12,13,14]. In these cases, it was calculated as the distance (with the Euclidean formula) between each cancelled mark and the next one, excluding perseverations. This serves as an organized path meant to minimize the distance between the newly cancelled item and the one which the participant will decide to mark next.■*Number of intersections, intersections rate, longest path.* These three indexes refer to the same construct, so they are placed together. The number of intersections is the sum of each intersection into the imaginary line that links each disk to the next one. The intersection rate is calculated averaging the number of intersections (that is, dividing it for the 15 segments) while the longest path is the highest number, for each sequence, of intersections-free lines. In other words, if a pattern contains no intersection, the longest path is 15. These indexes were used by several authors as a visual search organization measure [12,13,14]. Conceptually, a good quality sequence should not come back to the same spatial position as before; that is, the sequence should not intersect with itself.■*Global speed.* Global speed is calculated as the ratio between the within distance and the total time. Virtually, the lower the speed, the more disorganized the pattern should be.■*Best R (R_X and R_Y).* The vast majority of healthy participants cancelled items with a horizontal or vertical movement (that is, by rows or by columns) [17,18,20]. This was also common for the E-BTT (see [11] for more details). To address this, Mark and colleagues created the best R, the highest, in absolute value, among the two Pearson correlations between the coordinates (X and Y separately) and their order. The main limitation of this approach was that this index did not catch patterns other than orthogonal ones, for example, spiral paths. A further limitation is that choosing to use only the highest between two values makes it impossible to know which one is horizontal or vertical. We chose to use both correlation coefficients.■*Standardized angle.* As an integration to the information given by the best R, Dalmaijer [13] developed a standardized measure of the mean angle of the patterns’ segments. The angle between two points is calculated as the arcsin of the ratio between the vertical distance (the difference between the two Ys) between two points and their Euclidean distance. Then, each angle is standardized and averaged. The higher the standardized angle, the more efficient the pattern should be.■*Between distance from optimal sequences*. This last index was proposed recently based on the results shown in Argiuolo and colleagues [11]. It consisted of the “between distance”—or the distance between the corresponding disks of two distinct sequences—from sequences that could be considered optimal. They were created based on the fact that as for the E-BTT, the goal of the task was to place the disks inside the frame as evenly as possible. Therefore, the optimal disposition should occupy as much space as possible. The 16 disks’ coordinates were calculated by dividing the available area in 16 equal parts. The result was an optimal disposition of four per four disks; the only variable now was the sequence of disposition. Following the examples from cancellation tasks, we wondered whether the sequence by rows and columns was also applicable for the E-BTT. Indeed, the most frequent sequences in Argiuolo and colleagues’ [11] groups were the first two groups where a sawtooth was the final result. The sawtooth goes by rows and columns, similar to a commonly used cancellation path reported by Warren, Moore and Vogtle [18]. Therefore, we chose these two sequences as optima and calculated the BDs from them (Figure 2). The results were two indexes that gave an idea of how different each particular sequence was from the optimal ones. Of course, this choice had limitations; future research could take this into account and also consider other optimal sequences.


### 2.3. Data Analysis

Analyses were performed with the Jamovi software (Version 2.2.5.0; Jamovi Project, Sydney, Australia) [21].

In order to assess their structure, a principal component analysis (PCA) was conducted on the indexes with an orthogonal rotation (Varimax). In the analysis, the time of performance was discarded because, as in the table (see Table 2), its kurtosis was too high. In the explorative attempts to include performance time, a supplementary component appeared. This meant that this index saturated in a different component than the others. Between the number of intersections, intersection rate and longest path, the latter was preferred for its lower kurtosis. Best R was also not included because the two single correlations were included instead. This analysis was performed on standardized scores.

## 3. Results

### 3.1. Right/Left Asymmetry

As for the classic BTT, a cutoff difference of more than two cubes was considered a sign of neglect [1]. In our sample of healthy participants, 13% of participants had an asymmetrical configuration more skewed than two in absolute value, and only three out of 122 performed asymmetrically towards the right side. Plus, 29.5% of participants had a score more skewed than the cutoffs established by Facchin and colleagues [10] (−12.6% for the left side and +18.8% for the right side). Interestingly, as in the right/left difference, only few of them were asymmetrical towards the right; the vast majority were biased toward the left.

### 3.2. Indexes’ Structure

In Table 2, descriptive statistics of each index are shown.

Since there is a gender difference in spatial abilities, we also report a table divided for gender (see Table 3).

As already mentioned, we performed a principal component analysis (PCA). The sample adequacy was analyzed through the Kaiser–Meyer–Olkin measure, KMO = 0.671; Kaiser [22] stated that the minimum should be 0.5, and values between 0.5 and 0.7 could be considered mediocre. This, of course, was due to the small sample size. Singularly, each index’s KMO was above 0.5, the acceptable limit [23]. Bartlett’s test of sphericity was significative, χ^2^ (171) = 2254, *p* < 0.05.

From the scree plot (see Appendix A), six components had eigenvalues above the one (Kaiser’s criterion) and explained 82.5% of variance in combination. It is also true that the scree plot can be considered ambiguous since the inflexions lied in the third and fifth component. Therefore, based on parallel analysis, five components were retained rather than six. Parallel analysis [24] is a statistical technique that helps to decide how many components or factors retain in a PCA or an exploratory factor analysis. It is considered a better criterion than Kaiser’s in many situations [25]. Table 4 shows the factor loadings after rotation.

Results shows (Figure 3):A component classifiable as verticality (mean Y, verticality bias, U/D difference, first Y, distance vertical gradient) and a component of laterality (mean X, laterality bias, R/L difference).An “explored space/quality” component made up of within distance, total area, global speed and longest path.A component consisting of the distance from the first optimal sequence (OS1_BD), the distance lateral gradient (DLG) and the correlation between Y coordinates and their order (R_Y).Similarly, the distance from the second optimal sequence (OS2_BD), the correlation between X coordinates and their order (R_X), the angle (standardized angle) and the first X goes into the second component.

## 4. Discussion and Conclusions

In this study, we have applied several indexes to the Enhanced Baking Tray Task data in order to summarize possible measures that can be applied to this task.

As we expected, there was one verticality component (mean Y, verticality bias, U/D difference, first Y, distance vertical gradient) and one that described laterality (mean X, laterality bias, R/L difference). The fact that these two components were separated means that they probably refer to distinct aspects of peripersonal spatial exploration. We calculated the PCA on healthy participants to identify the dimensions to focus on both in research and clinical settings and to explore wider normative sets of data to be used in clinical settings.

Laterality and verticality are both two fundamental concepts in spatial cognition, and they were widely indagated.

The dimension of laterality has been generally investigated both in the literature on neglect [1,3,8,9,10] and in healthy subjects in the study of the so-called pseudoneglect [2,26,27,28,29,30]. The use of laterality indexes like the mean X, laterality bias or R/L difference could allow the assessment in spatial deficit cases since they are, by definition, the difficulty of paying attention to and being conscious of stimuli from the contralesional part of the space (usually, the left one) [31]. The study of the relative position of patients’ responses on the X-axis in several tasks (such as cancellation of items or line bisection) is obviously crucial for the neglect assessment and diagnosis.

In a recent study of pseudoneglect [2], assessed with the mean X and the first X coordinate, the results showed that both of these indexes were shifted leftward. Again, using such a measure also allowed the study of laterality in healthy subjects.

With regards to the vertical dimension, in right hemisphere stroke patients, perception of verticality is often impaired [32,33,34,35,36]. This impairment has been studied as an explanation of postural disorders in these patients, such as lateropulsion or pushing behavior [34]: altered verticality led patients to align their posture in a wrong way, tilted to the contralesional side. This impairment has been observed in neglect patients, and it is multimodal [33] because it concerns both the visual and the haptic modalities.

Secondly, vertical position was found to be associated with different abstract constructs, such as power, concreteness, valence, rationality/emotions and direction [37]. All these domains reveal that the two ends of the continuum (e.g., powerful/powerless, concrete/abstract, etc.) are connected and associated with one of the two dimensions of vertical space: up/down. For example, Shubert [38] found that people were quicker to recognize stimuli representing power (i.e., pictures labeled “master” or “servant”) when these appeared near the top of the screen and slower when the stimuli representing power were represented in the bottom. These results were interpreted as an index for the mental association strength: quicker reaction times meant stronger mental associations [37]. According to this interpretation, Meier and Robinson [39] found that participants were quicker to categorize positive words when these appeared at the top of the screen whereas the opposite was true for the negative words (shorter reaction time when they appeared at the bottom).

To the best of our knowledge, no study has addressed the issue of verticality in object disposition within peripersonal space, which the use of E-BTT could help doing.

Moreover, three out of five components are made up of indexes that express the quality of the sequence. Quality indexes inform “how” the disks have been placed, and they seem to have a multicomponent nature. On one hand, there is within distance, total area, global speed and longest path, components that address how the space was explored in terms of distance, time and organization. On the other hand, distances from optimal sequences and the correlation of the coordinates and their order formed another two components. Distance from an optimal sequence gives information about how much that single sequence is different from the sequence considered optimal. Since the two optimal sequences (see Figure 2) are orthogonal (that is, they go by column or row), it derives that a high distance from them means a high difference from the two principal orthogonal sequences, too. Two similar indexes, R_X and R_Y, catch the consistency of the search direction [40] and capture the orthogonal movement pattern [12]. Therefore, it is not surprising that they form two distinct components with the distance from optimal sequences.

What it is surprising is how the first X coordinates behave into the dimensions. Indeed, it results as part of the quality components instead of the laterality one. This could mean that its use is justified to express the quality of the pattern, rather than its spatiality.

Quality indexes have been mostly drawn from research on visual search organization, which focused on the pattern of cancellation, which is how participants/patients performed in an item cancellation task [12,13,14,41]. Clearly, the E-BTT is different from the cancellation task, inasmuch there are no targets but a completely empty space that should be organized by placing objects in it [3]. There are no correct or wrong configurations, and the patient/participant is free to place the disks however they like.

One of the strengths of the E-BTT is its ecological validity since the task involves a daily activity, such as baking cookies and placing dough on a tray. Ecological validity, indeed, regards the capacity of a task to generalize its results to settings that are different from the clinical or research one [42]. Besides, assessing the ecological validity of neuropsychological tests has become more and more important for its implication in research and rehabilitation [43].

The use of these indexes in the future may, therefore, allow the study of different dimensions of exploration in peripersonal space. Similar work was conducted by Dalmaijer [13] and colleagues, in that they implemented a series of measures into their software tool, Cancellation Tools. However, they only tested them on a small sample (*n* = 20), so they could not analyze their latent dimensionality.

The approach we used here can be useful to answer the question about what main components underlying these measures are; of course, these are preliminary results, but we can conclude that different latent dimensions exist, which refer to a spatiality dimension (see Figure 3) divided into laterality (mean X, laterality bias, R/L difference) and verticality (mean Y, verticality bias, U/D difference, first Y, distance vertical gradient). The other components are made up of indexes that express the quality of the pattern and that deal with the “how” and “how organized” the disks were placed by participants. Currently, we are working on a preliminary application of E-BTT in a clinical setting that has led to encouraging results and can be considered for the future application to teleneuropsychology [44].

A limitation to this study is the relatively small sample size (*n* = 122) that did not allow the establishment of norm scores. Future studies should address this issue by expanding the sample in order to confirm the indexes’ structure. Moreover, establishing norm scores could be helpful in distinguishing healthy participants from patients with different diagnoses. To this end, it will be important to include older participants, who are more likely to match stroke patients in age. We also aim at recruiting different population of patients, like right brain damage or posterior vs. frontal damage patients.

In conclusion, using these indexes in the E-BTT task could help to establish a reference point that can be used in diagnosis and in research as they measure the different aspects of visuo-spatial strategies in peripersonal space (laterality, verticality and quality).

## Figures and Tables

**Figure 1 brainsci-12-00401-f001:**
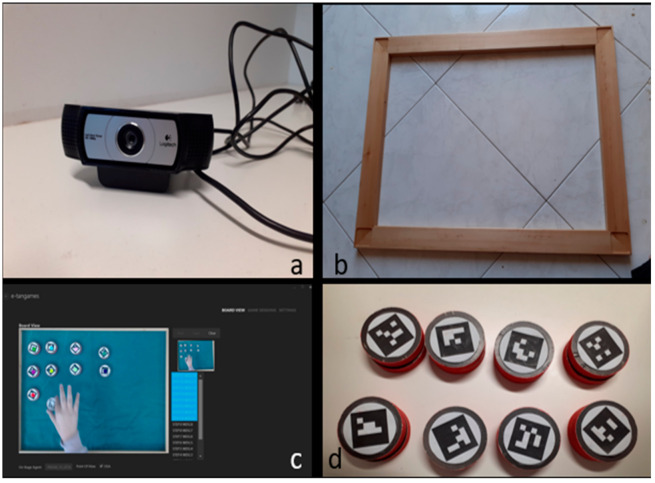
Some pictures of the E-BTT apparatus: (**a**) the Logitech C930e webcam camera (Newark, CA 94560, USA); (**b**) the wooden frame; (**c**) a screenshot from the board view of the software platform; (**d**) the 5 cm disks with ArUco markers on them.

**Figure 2 brainsci-12-00401-f002:**
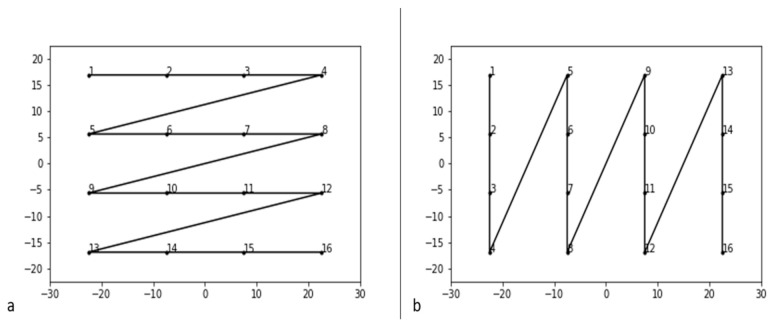
The two optimal sequences: (**a**) the first goes by rows (OS1); (**b**) the second goes by columns (OS2).

**Figure 3 brainsci-12-00401-f003:**
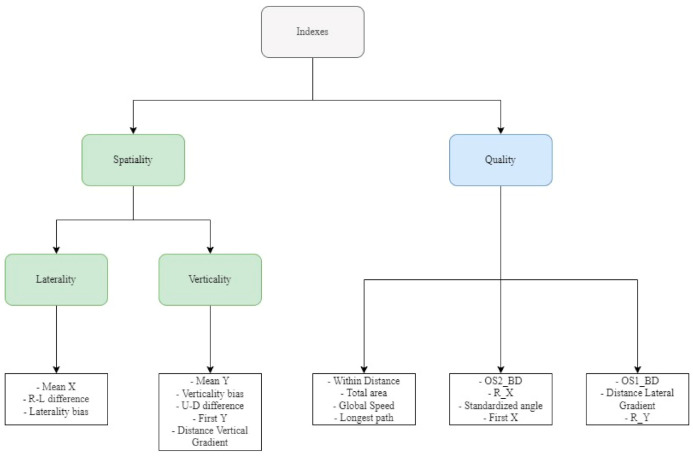
Indexes’ structure.

**Table 1 brainsci-12-00401-t001:** List of all the indexes.

Type	Name	What Measures	References
Spatial	Quadrant analysis (The quadrant analysis is included as an important measure of disks’ placement tendency even though it is not an index.)	1 × 4 chi-square on disks’ placement frequencies	Cerrato et al., 2019; Somma et al., 2020
Laterality	First X	First disk’s X coordinate	Somma et al., 2020
	R/L difference	Difference of disks placed on the right and disks placed on the left part	Tham & Tégner, 1996; Cerrato et al., 2020; Karnath, et al., 2002; Natale, et al., 2007
	Laterality bias	Ratio between R/L disks’ difference and their total number, in a percentage	Facchin et al., 2016; Cerrato et al., 2020
	Distance lateral gradient	Slope of the regression line, putting the distance between each disk and the next one as a dependent variable and the X coordinates as a predictor	Rabuffetti et al., 2012
	Mean X	Mean of the 16 X coordinates	Somma et al., 2020
Verticality (Verticality indexes are specular to laterality ones.)	First Y	First disk’s Y coordinate	Present Study
	U/D difference	Difference of disks placed on the top and disks placed on the bottom part	
	Verticality bias	Ratio between U/D disks’ difference and their total number, in a percentage	
	Distance vertical gradient	Slope of the regression line, putting the distance between each disk and the next one as a dependent variable and the Y coordinates as a predictor	
	Mean Y	Mean of the 16 Y coordinates	
Quality	Total area	Proportion of space occupied by the convex hull delimited by the disks	Cerrato et al., 2020
	Total time	Performance time in seconds from the first to the last disk	Dalmaijer et al., 2015; Rabuffetti et al., 2012
	Number of intersections	The number of time two distinct segments crossed each other	Mark et al., 2004; Woods & Mark, 2007
	Longest path	The highest number, for each sequence, of consecutive intersections-free lines	Rabuffetti et al., 2012;
	Intersection rate	The number of time two distinct segments crossed each other, divided by the number of total segments	Dalmaijer et al., 2015; Woods & Mark, 2007
	Best R	The highest, in absolute value, between the two Pearson’s correlation between coordinates and their order	Dalmaijer et al., 2015; Mark et al., 2004; Woods & Mark, 2007
	Standardized angles	Mean of the segments’ angles	Dalmaijer et al., 2015;
	Global speed	Ratio of within distance and total time	Dalmaijer et al., 2015; Rabuffetti et al., 2012
	Within distance	Sum of each disk and the next one’s distance	Dalmaijer et al., 2015; Argiuolo et al., 2021; Mark et al., 2004; Woods & Mark, 2007; Rabuffetti et al., 2012
	Optimal sequences between distance	Distances from two optimal configurations (rows and columns sawtooth)	Present Study

**Table 2 brainsci-12-00401-t002:** Descriptive statistics for the list of indexes. OS1_BD = optima sequence number 1_between distance; OS2_BD = optima sequence number 2_between distance; WD = within distance; DLG = distance lateral gradient; DVG = distance vertical gradient.

Index’s Name	Min	Max	Mean	Median	First Quartile	Third Quartile	SD	Asymmetry	Kurtosis
First X	−27.297	26.667	−16.013	−23.319	−25.174	−19.971	17.317	1.855	1.668
First Y	−19.333	19.393	4.129	14.743	−16.464	17.051	16.161	−0.569	−1.637
Mean X	−21.63	6.988	−1.363	−0.591	−2.44	0.27	3.515	−2.53	11.327
Mean Y	−17.557	16.403	−0.031	0.067	−0.932	1.762	5.749	−0.629	2.821
Total time	23	408	57.033	47.5	34	62.25	42.852	5.392	39.082
WD	83.176	520.463	260.645	242.577	205.419	297.708	91.714	0.84	0.886
OS1_BD	18.076	577.129	349.490	390.002	301.350	443.89	139.419	−1.104	0.396
OS2_BD	33.114	635.118	356.369	378.245	322.179	422.847	123.327	−0.895	0.893
R/L difference	−16	6	−0.82	0	−0.25	0	3.072	−3.169	13.54
U/D difference	−16	16	−0.016	0	0.00	2	6.259	−0.407	2.314
Laterality bias	−100	31.25	−7.018	0	−18.75	0	18.682	−1.723	6.449
Verticality bias	−100	100	−0.102	0	−6.25	18.75	38.376	−0.367	2.023
DLG	−0.653	0.525	−0.144	−0.068	−0.381	0.074	0.272	−0.234	−0.857
DVG	−2.565	5.233	−0.164	−0.073	−0.478	0.252	0.981	0.781	7.268
R_X	−0.971	0.982	0.323	0.263	0.092	0.704	0.485	−0.614	0.379
R_Y	−0.983	0.978	−0.145	−0.171	−0.887	0.290	0.643	0.239	−1.068
Best R	−0.983	0.982	0.069	0.249	−0.930	0.927	0.825	−0.167	−1.761
Standardized angle	0.978	1.023	1.002	1.001	0.998	1.004	0.01	0.12	1.179
Global speed	0.522	14.743	5.570	4.952	3.724	7.243	2.678	0.890	1.338
Intersection number	0	29	3.008	0	0	2.25	6.508	2.711	6.876
Intersection rate	0	1.933	0.201	0	0	0.15	0.434	2.711	6.876
Longest path	0	15	10.88	15	4	15	5.845	−0.854	−1.092
Total area	0.041	0.643	0.355	0.374	0.296	0.436	0.136	−0.492	0.025

**Table 3 brainsci-12-00401-t003:** Descriptive statistics for the list of indexes divided for gender. OS1_BD = optima sequence number 1_between distance; OS2_BD = optima sequence number 2_between distance; WD = within distance; DLG = distance lateral gradient; DVG = distance vertical gradient.

	Mean		Median		Standard Deviation	
Gender	Female	Male	Female	Male	Female	Male
FirstX	−17.081	−11.871	−23.275	−23.492	15.973	21.653
FirstY	4.671	2.026	14.75	13.311	15.928	17.211
Mean_X	−1.342	−1.444	−0.713	−0.417	3.098	4.885
Mean_Y	0.059	−0.379	0.021	0.568	5.453	6.896
Total time	55.567	62.72	46	53	45.067	33.042
WD	259.508	265.054	242.852	233.99	86.778	110.727
OS1_BD	335.368	404.282	376.106	409.524	143.052	110.498
OS2_BD	355.338	360.372	376.259	402.288	114.39	155.95
R/L difference	−0.887	−0.56	0	0	2.94	3.595
U/D difference	0.165	−0.72	0	0	6.032	7.162
Laterality bias	−7.023	−7	0	0	17.051	24.428
Verticality bias	0.258	−1.5	0	0	37.39	42.782
DLG	−0.156	−0.099	−0.076	−0.023	0.275	0.263
DVG	−0.171	−0.136	−0.11	−0.041	1.012	0.868
R_X	0.34	0.26	0.278	0.223	0.473	0.533
R_Y	−0.181	−0.005	−0.173	−0.168	0.647	0.62
Best R	0.027	0.236	0.214	0.646	0.835	0.779
Standardized angle	1.001	1.003	1.001	1.001	0.01	0.01
Global speed	5.668	5.187	5.281	4.661	2.565	3.104
Intersection number	2.897	3.44	0	0	6.262	7.512
Intersection rate	0.193	0.229	0	0	0.417	0.501
Longest path	10.948	10.6	15	15	5.86	5.895
Total area	0.356	0.352	0.374	0.379	0.132	0.154

**Table 4 brainsci-12-00401-t004:** Summary of the principal component analysis for the indexes (*n* = 122). Eigenvalues and uniqueness are reported. OS1_BD = optima sequence number 1_between distance (horizontal, see Figure 2); OS2_BD = optima sequence number 2_between distance (vertical, see Figure 2).

	Component					
	1	2	3	4	5	Uniqueness
Mean Y	**0.968**					0.053
Verticality bias	**0.958**					0.075
U/D difference	**0.953**					0.076
First Y	**0.549**	0.489		−0.455		0.247
Distance vertical gradient	**−0.409**			0.379		0.683
OS2_BD		**−0.954**				0.068
R_X		**0.829**				0.245
Standardized angle		**0.678**				0.493
First X		**−0.596**		0.367		0.409
Mean X			**0.959**			0.048
R/L difference			**0.935**			0.105
Laterality bias			**0.933**			0.101
OS1_BD				**0.92**		0.127
Distance lateral gradient				**0.841**		0.233
R_Y	−0.584			**0.696**		0.142
Within distance					**0.932**	0.105
Total area					**0.803**	0.300
Global speed					**0.665**	0.491
Longest path				−0.332	**−0.578**	0.478
**Eigenvalues**	3.68	2.88	2.85	2.72	2.39	
**% of variance**	19.4	15.1	15.0	14.3	12.6	

## Data Availability

The data in this study are available on request to the corresponding author.

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
