# Peer review of "Indexes for the E-Baking Tray Task: A Look on Laterality, Verticality and Quality of Exploration"

_brainsci, 2022, doi:10.3390/brainsci12030401_

Round 1
Reviewer 1 Report
1. Given that the E-Baking Tray Task is closely related to activities of daily living, establishing its ecological validity, the additional implication should be focused on in terms of its ecological validity.
2. As the spatial ability between males and females is found to be different, the authors need to conduct a sub-group(gender group) analysis of main findings
Reviewer 2 Report
Summary
In this study, the authors had healthy participants complete the enhanced version of the baking tray task in which they must place 16 disks evenly within a rectangular frame. A camera and specialized software allowed for automated extraction of a number of measures related to the organization and efficiency of the disk placement. Principal components analysis grouped the measures into components addressing laterality, verticality, and quality. This task and analysis have applications in clinical assessments of spatial neglect.
General Comments
Overall, I think this task and the identification of meaningful components of spatial exploration would be helpful in studying neglect patients and, thus, the study addresses an important topic. My main concerns are 1) the originality of the current study and how it advances the literature and 2) the organization of the manuscript.
For the first point, the only unique measures in this study appear to be verticality measures that mirror the typical laterality ones, and the distance from the optimal path. Yet the authors even note that these “optimal” paths are not the only possible solution and there is no correct sequence. Further, it is unclear to me how much value is added by the PCA and how using these components would provide an advantage over individual measures, which seems difficult to justify without data from neglect patients.
Secondly, I think the clarity of the manuscript could be much improved by careful attention to what is described in each section (see specific comments). In particular, the list of measures should be written more concisely.
Specific Comments
Lines 52-69 – This list feels too detailed for the introduction section. Relevant changes for the E-BTT could be described more broadly in paragraph form, and specifics described in the methods.
Line 78 – “The choice of centimeters…” The rest of the sentence seems to describe the advantage of changing the center from the lower right to the real center, not changing from pixels to cm.
Methods – There should be a description of the instructions given to participants. Also, were they allowed to move the disks after the initial placement?
Line 171/183/214/230/243/292 – The references to findings or conclusions of other studies should be removed or limited in the methods sections to present more concisely what the measures themselves are, with comparisons to previous work made in the discussion or intro.
Line 194/209 – These descriptions of the current data should be reported in the Results section.
Table 1 – It might make the table easier to read if “Type” is the first column, then “Name”.
Line 373 – At least this first paragraph about the PCA should be in the methods section to describe the analysis.
Figure 3 – I don’t think this figure is necessary to justify your choice of five components.
Table 3 – Please define in the caption what the values in the table for each component are, and the uniqueness value.
Figure 4 – As I understand it, your primary/final analysis selected only 5 components, so the total time box (6th component) should not be included here.
Line 417 – The first half of this paragraph simply repeats what you said previously and the second half should be in the discussion.
Line 508 – “establish patterns of strategies (effective vs ineffective)” – It’s unclear to me how this analysis assesses effective vs ineffective strategies. Generally, I think a stronger conclusion is needed to drive home what the advantage of the current analysis is.
